# Self-Supervised Node Classification with Strategy and Actively Selected Labeled Set

**DOI:** 10.3390/e25010030

**Published:** 2022-12-23

**Authors:** Yi Kang, Ke Liu, Zhiyuan Cao, Jiacai Zhang

**Affiliations:** School of Artificial Intelligence, Beijing Normal University, No. 19 Xinjiekouwai Street, Haidian District, Beijing 100875, China

**Keywords:** graph neural network, self-supervised learning, semi-supervised classification, automatic hyperparameter optimization, active learning

## Abstract

To alleviate the impact of insufficient labels in less-labeled classification problems, self-supervised learning improves the performance of graph neural networks (GNNs) by focusing on the information of unlabeled nodes. However, none of the existing self-supervised pretext tasks perform optimally on different datasets, and the choice of hyperparameters is also included when combining self-supervised and supervised tasks. To select the best-performing self-supervised pretext task for each dataset and optimize the hyperparameters with no expert experience needed, we propose a novel auto graph self-supervised learning framework and enhance this framework with a one-shot active learning method. Experimental results on three real world citation datasets show that training GNNs with automatically optimized pretext tasks can achieve or even surpass the classification accuracy obtained with manually designed pretext tasks. On this basis, compared with using randomly selected labeled nodes, using actively selected labeled nodes can further improve the classification performance of GNNs. Both the active selection and the automatic optimization contribute to semi-supervised node classification.

## 1. Introduction

Deep learning models, such as convolutional neural networks (CNNs), recurrent neural networks (RNNs), and autoencoders have shown excellent ability to extract information from data in an end-to-end way, and thus revolutionized many machine learning tasks such as visual recognition, signal analysis, and natural language understanding [1,2]. Most data used in the above tasks are represented in Euclidean space; however, observed samples in the real world are sometimes represented as graphs in non-Euclidean spaces [1]. At the same time, due to the high dimension features of graph data [3], deep learning methods play an important role in graph data analysis to extract features automatically in graph data. Inspired by CNNs and graph embeddings, graph neural networks (GNNs) are promising applications of deep learning that aggregate information from graph structures by introducing a layer-wise propagation rule for neural network models [1,4].

Due to the contribution from neighbors’ information, GNNs perform well on semi-supervised node classification tasks, as well. Nevertheless, since some labels are hard to obtain, sometimes training needs to be performed with few labels. If we rely heavily on label supervision information, problems such as over-fitting and weak robustness often occur when labeled nodes are scarce while the node features’ dimensions are excessively high. [5]. To reduce the reliance on labels, some auxiliary tasks called self-supervised pretext tasks are predesignated to make use of unlabeled data. Instead of using the given labels as supervision information, self-supervised pretext tasks use pseudo labels extracted from unlabeled data. Learning paradigms include self-supervised pretext tasks called self-supervised learning (SSL). SSL methods can be used in semi-supervised tasks where only a part of the samples is labeled, and can also be used in unsupervised tasks.

Corresponding to the self-supervised pretext task that mines information of unlabeled data, supervised tasks are called downstream tasks, and the downstream task in this study is the node classification task. To design an SSL framework that benefits downstream tasks, self-supervised strategies need to be considered, which involve the design of the pretest task and its integration with the training scheme. Good self-supervised strategies should be able to mine information complementary to supervised tasks. Therefore, designing the self-supervised strategy for a specific dataset needs to include consideration of both the data distribution and the existing label distribution.

There are a variety of self-supervised pretext tasks employed in graphs [5]. Generation-based pretexts use the input graphs as supervision and training models by minimizing the difference between input and reconstructed graphs [6]. Auxiliary property-based tasks use the properties of graphs as pseudo labels [7,8]. Contrast-based tasks aim to maximize the mutual information between augmentation views of the same sample [9,10].
(1)$$\begin{array}{c} \theta^{*},w_{s}^{*}=\underset{\theta ,w_{s}}{\arg\min}L_{s}\left(\theta ,w_{s}\right)\\ \theta^{**},w^{*}=\underset{\theta^{*},w}{\arg\min}L\left(\theta ,w\right) \end{array}$$
(2)$$\theta^{*},w^{*},w_{s}^{*}=\underset{\theta ,w,w_{s}}{\arg\min}L\left(\theta ,w\right)+\beta L_{s}\left(\theta ,w_{s}\right)$$

GNN training can be regarded as the process of seeking the optimal parameters that minimize the overall loss of the pretext task and the downstream task. The parameters of the GNN are denoted as θ, the linear transformation parameters of the pretext task and the downstream task are denoted as *w* and $w_{s}$, and the loss function of the pretext task and the downstream task are denoted as L(·,·) and $L_{s}\left(\cdot ,\cdot \right)$. Pretext task and downstream tasks can be trained separately in pre-training and fine-tuning as in (Equation 1) [8,11,12]. Introducing a hyperparameter β to represent the self-supervised loss’s scaling coefficient, these two tasks can also be trained jointly in a multitask manner, as in (Equation 2) [7,13,14]. The schematic diagrams of the two training schemes are shown in Figure 1.

Quite a few studies already give insight into the design of SSL strategies for specific data distributions. The performance of different pretext tasks was investigated in [7], including local structure information directed pretext tasks, global structure information directed pretext tasks, and attributed information directed pretext tasks. In addition to improve matching with labeled nodes, this study also proposes some pretext tasks incorporating the pieces of information of the labeled nodes. However, there is no method that is generally applicable to all data that performs well on all datasets. Moreover, it is time-consuming and laborious to determine the right strategy for a specific dataset. The work in [14] also focuses on how to incorporate self-supervised learning with GNNs to maximize the performance improvement on downstream tasks and the useful self-supervised pretext tasks for GNNs. As a result, three pretext tasks that the authors considered useful for GNNs were presented; however, the authors further illustrated that different models and datasets seemed to prefer different self-supervised tasks.

Although both [7,14] indicate that combining self-supervised pretext tasks and downstream tasks in a multi-task way has certain advantages over the pre-training and fine-tuning two-step mode in both semi-supervised classification performance and framework complexity, the value of scaling coefficient hyperparameter β still needs manual optimization.

Finding the adapted pretext task and hyperparameter manually often requires professional experience and may take much time. To reduce the reliance on priors and provide a more accurate choice, this study proposed a differentiation-based self-supervised strategy automatic optimization method.

In semi-supervised tasks, information from unlabeled data has been intensively studied. Meanwhile, manual annotation also makes a great contribution to performance improvement for semi-supervised learning. Given the labeling cost, selecting the valuable labeled data and measuring its contribution is a general problem in semi-supervised learning. As a branch of machine learning, active learning is mainly designed for a scenario with fewer data labels or higher costs of labeling. The model in active learning interacts with experts to determine the data label, and to make the model use less labeled data to obtain better performance [15]. A one-shot active learning method was introduced to work with our auto-optimized graph self-supervised learning (AGS) framework, and we called the whole framework the active auto graph self-supervised learning (AAGS) framework.

## 2. Related Works

Relevant works include studies of data-adapted graph self-supervised learning, auto hyperparameter optimization, and active learning for graphs.

### 2.1. Data Adapted Graph Self-Supervised Learning

Current self-supervised pretext tasks depend on specific task environments, and face challenges to generalize to other tasks. To improve the generalization performance, hybrid pretext tasks take advantage of different pretext tasks by combining them. GPT-GNN [16] uses an attribute generation task followed by an edge generation task to obtain intrinsic dependencies between node attributes and graph structure. The method in [17] investigates the joint training of downstream tasks and the combination of three self-supervised pretext tasks: node feature auto encoding task, corrupted feature reconstruction task, and corrupted embedding reconstruction task. The results show that some of the combinations of multiple pretext tasks fail to improve the performance, compared with one pretext task alone.

The main challenge of hybrid method design is assigning the weight of each pretext task. Faced with this challenge, Jin et al. proposed a novel index Pseudo-Homophily to measure the quality of combinations of pretext tasks, and determine the weights of pretext tasks automatically based on evolutionary algorithm and differentiable search [18]. The method mentioned above was used in unsupervised situations but is not suitable for semi-supervised tasks. Measures of the quality of pretext tasks through manually preset metrics will increase reliance on professional knowledge, and as the optimization process of pretext task weights is predefined before learning from labeled data, the complementarity of pretext and downstream tasks was not considered. In addition, using multiple pretext tasks will increase the dimension of the parameter space. Considering redundancy between pretext tasks and the sensitivity of small datasets to parameters, sometimes more pretext tasks do not lead to a better result. The results on three datasets in [17] show that on two of the datasets, a single pretext task assistant graph convolutional network (GCN) achieves the highest classification accuracy in downstream tasks. On the remaining dataset, combining two pretext tasks can obtain the best results. In contrast, the method using all pretext tasks did not achieve optimal results on all of the test data.

### 2.2. Auto Machine Learning

Selecting the optimal pretext task and optimizing the scaling coefficient hyperparameter automatically can be unified as an auto machine learning (autoML) problem that maximizes the performance of learning tools with limited human assistance and limited computational budget [19]. Mainstream optimized approaches used in auto machine learning include random search [20], grid search [21], Bayesian optimization [22], evolutionary methods [23], reinforcement learning methods [24], and gradient-based methods.

Since the search direction is guided by the gradient, gradient-based methods have efficiency advantages over other algorithms and offer more accurate information [19]. However, the premise of optimizing hyperparameters by gradient descent is that the loss function is differentiable. Based on the continuous relaxation of architectural representation, DARTS [25] converts the discrete search space of the network structure search into a continuous differentiable space, making it possible to use gradient descent to efficiently search for architectures. The DARTS algorithm can search for network structures with excellent performance, and the search speed is orders of magnitude faster than that of the non-differentiable algorithm.

Most autoML optimization problems optimize parameters by minimizing loss on the training set and optimize hyperparameters by minimizing loss on the validation set [19,26]; the loss functions used in the training loss and validation loss are the same. However, auto optimization in few label self-supervised learning studies faced some particular problems to be solved. First, only a small fraction of the data have labels, and labeled set is split into training and validation sets, which then result in fewer samples in each set. Second, in self-supervised learning, loss functions used in training and validation are different. Parameters and hyperparameters related to self-supervised pretext tasks exist only in training loss, which brings inconvenience to calculating the derivative of validation over them.

### 2.3. Active Learning for Graphs

Traditional active learning algorithms can be divided into three categories based on the query strategy [27]. Heterogeneity based algorithms select the most unknown instances, including uncertainty sampling [28], query-by-committee [29], and expected model Change [30]. Performance-based algorithms select the instances that make the model performs better, including expected error reduction [31] and expected variance reduction [32]. Representative-based algorithms select the instance that can represent the underlying distribution [33]. Active graph embedding as the representative work on the graph active learning framework [34] linearly combines the properties of the nodes and the entropy of the classifier into its discriminant function. Afterwards, many other graph active learning methods have emerged, such as meta-learning-based active learning [35], teacher–student model-based active learning [36], adversarial-based active learning [37], etc.

The algorithms described above all have iterative processes that retrain the machine learning model after selecting samples, which greatly increases the computational complexity. To improve the computing efficiency of active learning, one-shot active learning methods, such as Weisfeiler–Lehman sampling [38], selects all samples to be labeled based on the structural role of nodes before model training.

Both self-supervised learning and active learning aim at reducing labeling effort. Studies in the computer vision and natural language processing fields have shown that the self-supervised learning method is more effective than the active learning method in reducing the labeling effort, and active learning can offer additional benefits to self-supervised learning [39,40]. However, the self-supervised learning framework enhanced by active learning for graph data still needs to be further explored.

## 3. Methods

The schematic diagram of AAGS is shown in Figure 2. This section presents methods involved in the AAGS framework. First, we formalize the automatic graph self-supervised learning problem, then introduce hyperparameter optimization algorithms for solving the above problem. Finally, we describe how to incorporate active learning methods to assist the automatic graph self-supervised framework.

### 3.1. Problem Statement for Auto Graph Self-Supervised Learning

A graph *G* is denoted as G={V,E}, where $V=\{v_{1},v_{2},v_{3},\ldots ,v_{n}\}$ is the vertex(node) set of graph *G* and $E=\{e_{1},e_{2},e_{3},\ldots ,e_{m}\}$ is the edge set. In this study, we focus on binary undirected attribute graphs, whose edge weight takes the value of either 0 (unconnected) or 1 (connected). Each vertex has a k-dimensional attribute, and all node attributes are stacked to form attribute matrix $X=\left(x_{ij}\right)\in R^{n\times k}$.

When only a few nodes in the graph have labels, we hope that data-driven automatic optimization algorithms can automatically design self-supervised strategies that can assist supervised graph network classification. Suppose there are *c* candidate self-supervised pretext tasks, the set of these pretext tasks is denoted as $P=\{p_{1},p_{2},\ldots ,p_{c}\}$. All the candidate pretext tasks are trained with downstream node classification tasks in a multi-task way, and the set of self-supervised loss scaling coefficients is denoted as $B=\{\beta_{1},\beta_{2},\ldots ,\beta_{c}\}$. The self-supervised framework of each $p_{i}$ is illustrated below: Graph network with parameters θ maps *G* into a low-dimensional space to obtain graph embedding. Afterward, linear transformations for the pretext task and downstream task are performed on the graph embedding. The downstream task linear parameters are written as *w*, and pretext task linear parameters as $w_{s}^{i}\in W_{s}=\left\{w_{s}^{1},w_{s}^{2},\ldots ,w_{s}^{c}\right\}$ (s: short for self). Downstream classification task loss can be expressed as a function of model parameters (θ, *w*) and labeled node set $N_{train}^{l}$: $L_{c}\left(\theta ,w,N_{train}^{l}\right)$. The self-supervised pretext task loss can be expressed as a function of model parameters (θ, $w_{s}$) and unlabeled node set $N_{train}^{n}$: $L_{s}^{i}\left(\theta ,w_{s}^{i}\right)$. The full loss is a linear combination of the downstream task loss and the pretext task loss weighted by the scaling coefficient, which is expressed as (Equation 3). The model was trained by minimizing it on the training set.
(3)$$L_{f}^{i}=L_{c}\left(\theta ,w\right)+\beta_{i}L_{s}^{i}\left(\theta ,w_{s}^{i}\right)$$

The purpose of auto-optimized self-supervised learning is to select $p_{i}$ and its corresponding $\beta_{i}$ which can assist the GNN in achieving the best performance in the downstream classification task. The performance of the GNN is monitored by the classification loss on the validation set.

We relax the pretext task selection problem as a soft classification problem to construct the continuous search space so that it can be solved by differential-based optimization methods. The sum of each $l_{f}^{i}$ weighted by its probability $\lambda_{i}\in \Lambda =\left\{\lambda_{1},\lambda_{2},\ldots ,\lambda_{c}\right\}$($\sum_{i=1}^{c}\lambda_{i}=1$) constitutes the mean full loss. The mean full losses on the training and validation sets can be expressed as (Equation 4) and (Equation 5).
(4)$$L_{t}\left(\Lambda ,B,\theta ,w,W_{s},N_{train}\right)=\sum_{i=1}^{c}\lambda_{i}\left(L_{c}\left(\theta ,w,N_{train}^{l}\right)+\beta_{i}L_{s}^{i}\left(\theta ,w_{s}^{i},N_{train}^{u}\right)\right)$$
(5)$$L_{v}\left(\Lambda ,\theta ,w,N_{val}\right)=\sum_{i=1}^{c}\lambda_{i}L_{c}\left(\theta ,w,N_{val}\right)$$

The optimization goal is to obtain such hyperparameters: with them, parameters optimized on the training set can also minimize the loss on the validation set, which can be formulated as the following functions:(6)$$\min L_{v}\left(\Lambda ,\theta^{*}\left(\Lambda ,B\right),w^{*}\left(\Lambda ,B\right),N_{val}\right)$$
(7)$$s.t.\left(\theta^{*},w^{*},W_{s}^{*}\right)=\underset{\theta ,w,W_{s}}{\arg\min}L_{t}\left(\Lambda ,B,\theta ,w,w_{s},N_{train}\right)$$

Since the variables in (Equation 6) are constrained to be the optimal solution of (Equation 7), this is a bilateral optimization problem.

### 3.2. Hyperparameter Optimization

To solve the bilateral problem as shown in (Equation 7) and (Equation 6), a second-order approximate differential-based optimization method was proposed in this study. Parameters $\left(\theta ,w,w_{s}\right)$ and hyperparameters (β,λ) are optimized according to the gradient alternately; that is, while optimizing parameters, hyperparameters are fixed, and vice versa. The step sizes of the gradient descent on the training set and validation set are denoted as $\eta_{t}$ and $\eta_{v}$. At the *n*th iteration, $\left(\theta^{n},w^{n},W_{s}^{n}\right)$ can be updated from:(8)$$\;\begin{array}{cc} \left(\theta^{n},w^{n},W_{s}^{n}\right) & =(\theta^{n-1},w^{n-1},W_{s}^{n-1})\\ -\eta_{t}\cdot \underset{\theta ,w,W_{s}}{▽}L_{t}(B^{n-1}, & \Lambda^{n-1},w^{n-1},\theta^{n-1},W_{s}^{n-1},N_{train}) \end{array}$$

Likewise, $\left(B^{n},\Lambda^{n}\right)$ can be updated from:(9)$$\;\left(B^{n},\Lambda^{n}\right)=\left(B^{n-1},\Lambda^{n-1}\right)-\eta_{v}\cdot \underset{B,\Lambda }{▽}L_{v}\left(\theta^{n},w^{n},\Lambda^{n-1},N_{val}\right)$$

As there is no direct relationship between $L_{v}\left(\theta^{n},w^{n},\Lambda^{n-1},N_{val}\right)$ and $B^{n-1}$, a bridge was built between them through (Equation 10).
(10)$$\left(\theta^{n},w^{n}\right)=\left(\theta^{n-1},w^{n-1}\right)-\eta_{t}\cdot \underset{\theta ,w}{▽}L_{t}\left(B^{n-1},\Lambda^{n-1},w^{n-1},\theta^{n-1}\right)$$

Substituting (Equation 10) into (Equation 9), second-order optimization can be performed as in (Equation 11):(11)$$\;\begin{array}{cc} \underset{B,\Lambda }{▽}L_{v}\left(\theta^{n},w^{n},\Lambda^{n-1},N_{val}\right)= & \underset{B,\Lambda }{▽}L_{v}((\left(\theta^{n-1},w^{n-1}\right)\\ -\eta_{t}\cdot \underset{\theta ,w}{▽}L_{t}(B^{n-1},\Lambda^{n-1}, & w^{n-1},\theta^{n-1})),\Lambda^{n-1}) \end{array}$$

For the convenience of calculation, (Equation 11) is expanded into (Equation 12) according to the chain rule:(12)$$\;\begin{array}{cc} \underset{B,\Lambda }{▽}L_{v}\left(\theta^{n},w^{n},\Lambda^{n-1},N_{val}\right)= & \underset{\Lambda }{▽}L_{v}\left(B^{n-1},\Lambda^{n-1},w^{n},\theta^{n},w_{s}^{n}\right)\\ -\underset{\Lambda }{▽}\underset{\theta ,w}{▽}L_{v}\left(\theta^{n},w^{n}\right)\cdot \eta_{t}\underset{B,\Lambda }{▽}\underset{\theta ,w}{▽}L_{t}( & B^{n-1},\Lambda^{n-1},w^{n-1},\theta^{n-1},W_{s}^{n-1}) \end{array}$$

The second term of (Equation 12) can be approximated as:(13)$$\;\frac{\eta_{t}}{2\epsilon }\left(\underset{B,\Lambda }{▽}L_{t}\left(B^{n-1},\Lambda^{n-1},w^{+},\theta^{+},W_{s}^{n-1}\right)-\underset{B,\Lambda }{▽}L_{t}\left(B^{n-1},\Lambda^{n-1},w^{-},\theta^{-},W_{s}^{n-1}\right)\right)$$
where $\left(w^{\pm },\theta^{\pm }\right)=\left(w^{n-1},\theta^{n-1}\right)\pm \epsilon \underset{\theta ,w}{▽}L_{v}\left(\theta^{n},w^{n}\right)$.

Thus far, the differentiation of $L_{v}\left(\theta^{n},w^{n},\Lambda^{n-1},N_{val}\right)$ with respect to $\left(B^{n-1},\Lambda^{n-1}\right)$ can be approximated as (Equation 14), and at the *n*th iteration, $\left(B^{n},\Lambda^{n}\right)$ can be obtained from (Equation 15).
(14)$$\;\underset{\Lambda }{▽}L_{v}\left(B^{n-1},\Lambda^{n-1},w^{n},\theta^{n},W_{s}^{n}\right))-\left(13\right)$$
(15)$$\left(B^{n},\Lambda^{n}\right)=\left(B^{n-1},\Lambda^{n-1}\right)-\eta_{v}\cdot \left(14\right)$$

The pseudocode of hyperparameter optimization is given in Algorithm 1. After hyperparameter optimization, the pretext task corresponding to the highest probability will be selected, which is denoted as $p_{s}$, s=argmaxΛ.

**Algorithm 1** Auto Graph Self-supervised strategy optimization
**Input:**

G={V,E}

 Parameters: hyperparameter optimization iterations $N_{select}$, β fine-tuning iterations $N_{\beta }$, labeled set L Initialization: $\theta^{0}$, $w^{0}$, $W_{s}^{0}$, $B^{0}$, $\Lambda^{0}$ divide L into 3 similar parts, $L_{0}\cup L_{1}\cup L_{2}=L$ **while**
$l\in \{L_{0},L_{1},L_{2}\}$
**do**    Validation set: *l*, training set: L∖l    **repeat**
      n←n−1      Update $\left(\theta^{n},w^{n},w_{s}^{n}\right)$ based on Equation (Equation 8)
      Update $\left(B^{n},\Lambda^{n}\right)$ based on Equation (Equation 15)
    **until** $n\geq N_{select}$ **end while** i←argmaxΛ, set $\lambda_{i}\leftarrow 1$, $\lambda_{n}\leftarrow 0\left(n\neq i\right)$ **while**
$l\in \{L_{0},L_{1},L_{2}\}$
**do**    Validation set: *l*, training set: L∖l    **repeat**
      n←n−1      Update $\left(\theta^{n},w^{n},w_{s}^{n}\right)$ based on Equation (Equation 8)      Update $B^{n}$ based on Equation (Equation 15)    **until** $n\geq N_{\beta }$ **end while**
**Output:**

$\theta ,w,W_{s},B,\Lambda$



### 3.3. Select Nodes to Be Labeled Actively

To further improve the performance of the model under certain labeling costs, using active learning methods enhance the auto self-supervised learning framework. In this study, data-driven self-supervised strategy optimization is based on the distribution of labeled and unlabeled data. To avoid data distribution changes in the optimization process, a one-shot active learning method, Weisfeiler–Lehman sampling [38], is adopted, which selects all nodes to be labeled before model optimization. Batch size *k*, exploration budget EB, and the label budget LB need to be preset. Weisfeiler–Lehman sampling randomly picks *k* seed nodes from unlabeled nodes to generate its EB hop neighborhood subgraph and then used the Weisfeiler–Lehman algorithm [41] to give each subgraph its final color labels. Finally, the unlabeled nodes with the top rank color label(*k* nodes) are chosen to label. Sampling will repeat until reaching the labeling budget. The pseudocode of Weisfeiler–Lehman sampling is present in Algorithm 2.
**Algorithm 2** Weisfeiler–Lehman sampling [38]**Input:**G={V,E}
 Parameters: batch size *k*, labeling budget LB, exploration budget EB
 Initialization: test set *T*, labeled set L=∅, unlabeled set U=V∖T
 **while**
|U|<LB
**do**    S← Pick *k* random seed nodes from U
    **while** v∈S **do**
      $G_{s}\leftarrow$EB hop neighborhood subgraph of *v*
      R← Weisfeiler–Lehman (Gs)      R←R∩U      i←argmaxR, $n_{i}$ was labeled by oracles      $L\leftarrow n_{i}\cup L$, $U\leftarrow U\setminus n_{i}$    **end while** **end while**


## 4. Experimental Results

### 4.1. Datasets and Experiment Settings

We evaluated the performance of the AAGS framework on three benchmark citation datasets: Cora, Citeseer, and PubMed. These three datasets are all composed of a single binary undirected graph [42], and more specific information of them is shown in Table 1. Outliers in all graphs were pre-removed in this study.

A two-layer GCN was used as the backbone GNN model, and eight pretext tasks (Edge Mask, Distance to Cluster, Pairwise Attribute Similarity, Distance to Labeled, ICA Context Label, LP Context Label, Combined Context Label, and Node Property) mentioned in [7] were used as candidate pretext tasks. The labeled set was divided into three parts with similar sizes: one part is taken each time as the validation set, and the remaining parts are used as the training set. We performed hyperparameter optimization for 50 epochs on each validation set and selected the best performing epoch on the validation set to obtain the optimization result. After the selected pretext task was fixed, we fine-tuned β for two epochs. When all hyperparameters were settled, the self-supervised model was trained for 200 epochs on the full labeled set. Then, we assessed the performance of the automatic self-supervised learning framework by node classification accuracy on the test set. To mitigate the effects of random initialization, the automatic self-supervised learning framework ran five times with the same settings. We used the mean accuracy to present the results.

To investigate whether our automatic optimization algorithm could provide a self-supervised strategy suitable for the specific dataset we designed two rounds of experiments. Backbone GCN and its integration with each candidate pretext task were adopted as comparison methods, and the value of the scaling coefficient hyperparameter for each candidate pretext task was set according to the optimal value reported in [7]. In both rounds of experiments, the AGS framework and the comparison methods adopt the same labeled set partitioning scheme. In the first round, we followed the labeled set partition in [7], where the labeled set was composed of the first 20 nodes of every class. In the second round, we randomly selected 6, 12, 18, 24, and 30 nodes per class as the labeled sets to investigate the performance of automatic self-supervised strategies on different sizes of the labeled set.

After these two experiments, to verify the enhancement of active learning to the AGS framework, we compared the results of the automatic self-supervised framework on randomly selected and actively selected labeled sets. The labeling budget was set as 6n, 12n, 18n, 20n, 24n, and 30n, where *n* denotes the classes of nodes in datasets (Cora: 7, Citeseer: 6, PubMed: 3). The total number of labeled nodes is the same as the previous round of experiments, but neither the nodes selected actively nor selected randomly can be guaranteed to be equivalent for all classes. All experiments were tested on the same 1000-node test set.

### 4.2. Results

Table 2 shows the performances of all candidate pretext tasks and our auto-optimized self-supervised strategy. According to the table, the auto-optimized strategies achieve better results than the manually designed strategies on all three datasets.

The result of auto-optimized self-supervised strategies on different sizes of the labeled set is shown in Figure 3. The abscissa represents the number of labeled nodes selected for each class, and the ordinate represents the node classification accuracy on the test set. As a comparison, the blue line represents the performance of the 2-layer GCN without self-supervised tasks. The orange line represents the mean performance of all manually designed strategies. Manually designed strategies used in the first round of experiments are also used here. We can see in Figure 3 that the GCN with auto-optimized strategies outperforms the backbone GCN, and outperforms the mean performance of all manually designed strategies in most instances. In contrast, sometimes the mean performance of all manually designed strategies are only slightly better than the backbone GCN, and sometimes they even worsen.

The result of the automatic self-supervised learning framework on randomly and actively select labeled sets are shown in Figure 4. The orange line represents the performance of automatic self-supervised learning with actively selected labeled nodes, and the blue one represents the performance with randomly selected labeled nodes. The shadow on both sides of each line represents the standard deviation. The fully supervised results of the backbone GCN are plotted as a tomato red dashed line as the upper bound for self-supervised learning. On all three datasets, actively selected nodes provide more stable results than randomly selected nodes. The combination of active learning significantly improves the performance of the automatic self-supervised learning framework on datasets Cora and Citeseer. The comparison with the fully supervised result shows that the AAGS framework makes it possible to obtain classification results similar to fully supervised classification (Cora: 98.90%, Citeseer: 97.19%, PubMed: 97.86%) with just a small part of the data that were labeled (Cora: 7.98%, Citeseer: 5.53%, PubMed: 0.49%).

## 5. Discussion

To utilize supervised and self-supervised information effectively in a semi-supervised node classification task, this study proposes a novel automatically optimized self-supervised learning framework and enhances the framework by actively selecting a labeled set. The experimental results show that active selection and automatic optimization are both contributed to semi-supervised node classification.

We visualized the distribution of self-supervised policies and corresponding scaling coefficient values obtained by auto-optimization selection to further discuss the reasons why automatic optimization strategies could outperform manually designed strategies. The distribution of selected pretext tasks under different settings is shown in Figure 5. EM, DC, PD, D2L, ICL, LCL, CCL, and NP are the abbreviations for Edge Mask, Distance to Cluster, Pairwise Distance, Distance to Labeled, ICA Context Label, LP Context Label, Combined Context Label, and Node Property, respectively. From top to bottom, each row shows results with 6, 12, 18, 24, and 30 labeled nodes for each class. Three columns from left to right correspond to the results on the Cora, Citeseer and PubMed datasets. Under the same setting, the selected pretext text is concentrated in 1–3 pretext texts, and whether the change of dataset or the change of the size of the labeled set will affect the selection result. Figure 6 reflects the distribution of optimized beta values under different settings. As the labeled information increases, the reliance on the self-supervised pretext tasks decreases. On the Cora and Citeseer datasets, as the number of labeled nodes increases, the value of beta tends to decrease.

According to the results in the above two figures, we infer that auto-optimized self-supervised strategies can help GCN models achieve better results than the strategies manually designed for the following reasons: First, the automatic optimization algorithm can help different datasets to select matching pretext tasks. Second, the differentiation-based method can find some scale coefficient (β) values that are not traversed during manual design. In addition, we think it is also helpful to select a pretext task from multiple candidate pretext tasks instead of combining them all. Figure 3 shows that sometimes the average SSL results are only slightly better than the backbone GCN and sometimes even worsen. This result supports our point of view that not all pretext tasks will have a positive effect.

To intuitively display the feature extraction results of the AAGS framework and the selection results of the active learning, the t-SNE [43] dimensionality reduction visualization results of the Cora dataset is shown in Figure 7. In this figure, we can see that AAGS effectively transforms the nodes mixed in the original space into separable point clusters. Furthermore, we find that the nodes selected by active learning can reflect the overall node distribution to some extent. To quantitatively display the distribution of actively picked nodes, a heatmap was plotted with the proportion of each type of node shown in Figure 8. Each column of the heatmap represents one labeled set’s distribution, and the rightmost column represents the distribution of all nodes. It can be seen that the distribution of actively selected nodes is similar to the overall node distribution. On the PubMed dataset, active selection no longer brings additional help to the classification task as the number of labeled samples increases. This may be because there are few classes of nodes in the PubMed dataset, and the randomly selected nodes already contain a sufficient number of representative nodes when the labeled sample size reaches 24n and 30n.

On the basis of this study, extending the automatically optimized self-supervised learning framework to different kinds of graph datasets and combining network structure optimization with this framework can be further explored in the future.

## Figures and Tables

**Figure 1 entropy-25-00030-f001:**
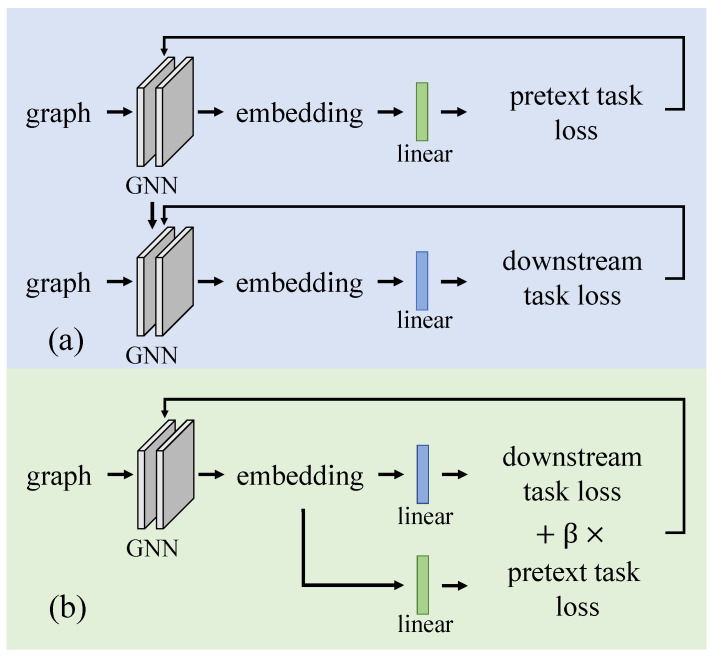
Two training schemes to integrate pretext task and downstream task. (**a**) Pre-training GNN on self-supervised pretext task at first and then fine-tuning GNN with the downstream task based on the pre-trained GNN. (**b**) The pretext task and the downstream task are trained jointly in a multi-task manner: in each iteration, the parameters of GNN and linear layers are updated according to the full loss, which is the sum of the β weighted pretext loss and the downstream task loss.

**Figure 2 entropy-25-00030-f002:**
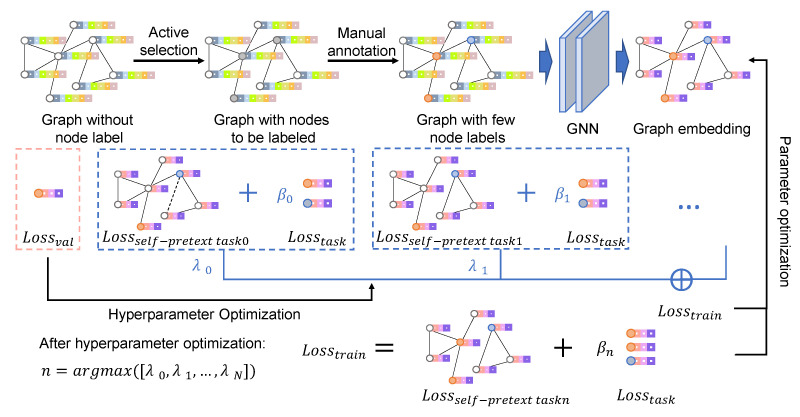
Schematic diagram of AAGS: Actively select the nodes to be labeled based on the labeling budget and the Weisfeiler–Lehman sampling algorithm. Manually annotate selected nodes, take one-third of the labeled nodes as the validation set, and the rest labeled nodes as the training set. Parameters and hyperparameters are alternately optimized: parameters are optimized by minimizing the loss on the training set, and hyperparameters are optimized by minimizing the loss on the validation set. After optimization iterations, the pretext task with the largest lambda value is selected to form the final self-supervised strategy.

**Figure 3 entropy-25-00030-f003:**
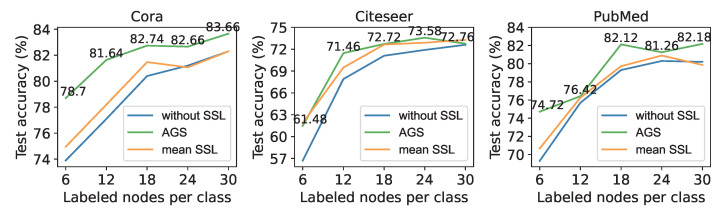
Performance of auto-optimized self-supervised learning with different amounts of labeled nodes.

**Figure 4 entropy-25-00030-f004:**
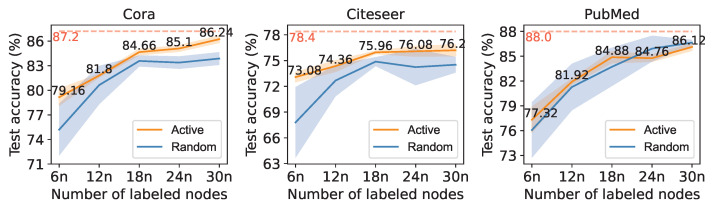
Performance of auto-optimized self-supervised learning with actively selected labeled nodes. *n* denotes the classes of nodes in a dataset (Cora: 7, Citeseer: 6, PubMed: 3).

**Figure 5 entropy-25-00030-f005:**
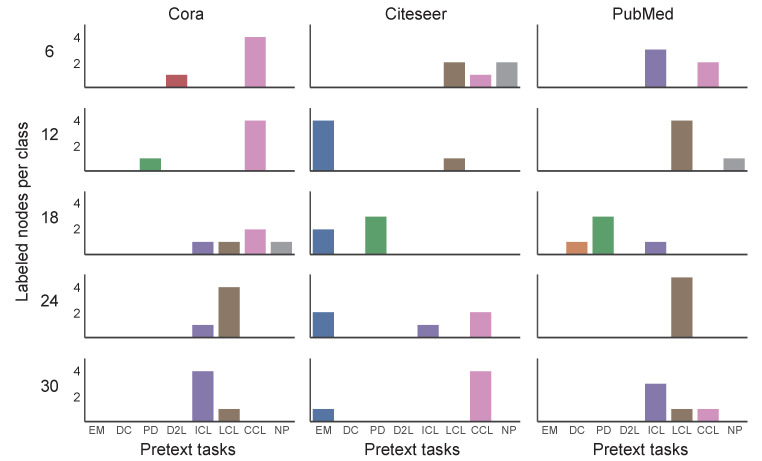
The distribution map of the pretext tasks selected by the automatic optimization algorithm.

**Figure 6 entropy-25-00030-f006:**
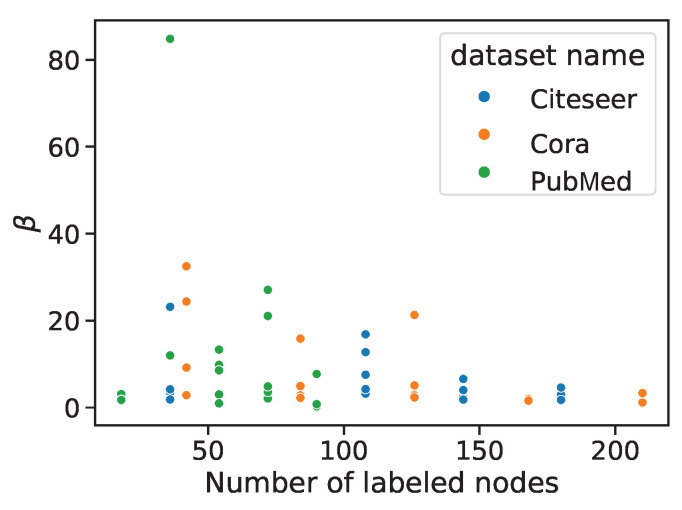
The distribution of values of β obtained by automatic optimization.

**Figure 7 entropy-25-00030-f007:**
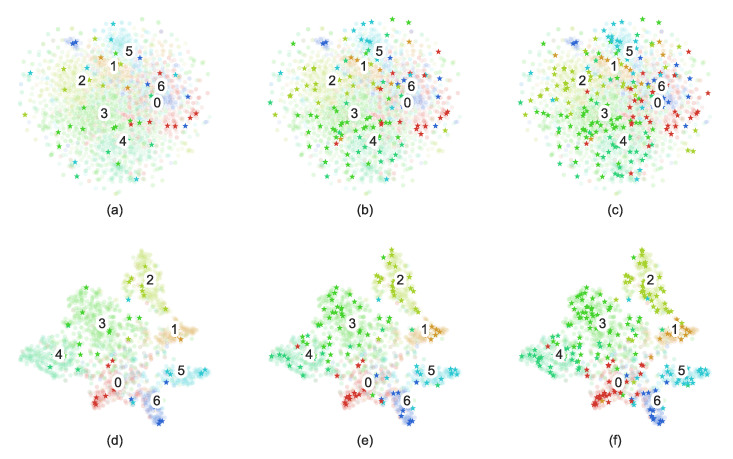
Visualization results on the Cora dataset using the t-SNE method: graph (**a**–**c**) and graph embedding (**d**–**f**). Translucent dots represent unlabeled nodes, stars represent labeled nodes, and the numbers represent the class numbers of the nodes. The labeling budgets of the first, second, and third columns are 6n, 18n, and 30n, respectively.

**Figure 8 entropy-25-00030-f008:**
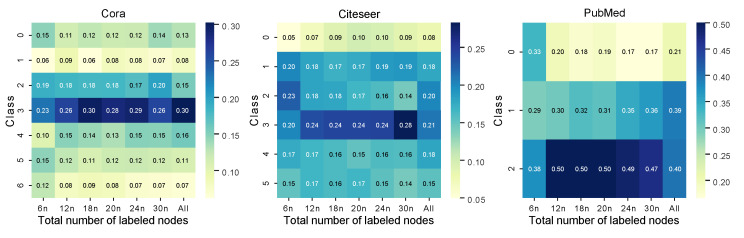
Frequency distribution of actively selected nodes over different classes, where *n* denotes the classes of nodes in a dataset (Cora: 7, Citeseer: 6, PubMed: 3).

**Table 1 entropy-25-00030-t001:** Dataset statistics.

Dataset	Graph	Nodes	Edges	Classes	Features
Cora	1	2708	5429	7	1433
Citeseer	1	3312	4732	6	3703
PubMed	1	19,717	44,338	3	500

**Table 2 entropy-25-00030-t002:** Node classification performance of integrating SSL into GCN.

Self-Supervised Tasks	Dataset		
	Cora	Citeseer	PubMed
GCN backbone	81.52 ± 0.43%	71.94 ± 0.22%	79.46 ± 0.30%
EdgeMask	82.04 ± 0.88%	71.36 ± 1.06%	79.34 ± 0.54%
Distance2Cluster	83.22 ± 0.44%	71.60 ± 0.37%	79.02 ± 0.38%
PairwiseAttrSim	82.14 ± 0.52%	71.92 ± 0.12%	79.52 ± 0.31%
Distance2Labeled	83.36 ± 0.72%	71.52 ± 0.49%	79.10 ± 0.28%
ICAContextLabel	83.54 ± 0.39%	72.70 ± 0.53%	82.38 ± 0.18%
LPContextLabel	81.38 ± 0.60%	72.44 ± 0.46%	79.86 ± 0.27%
CombinedContextLabel	83.32 ± 0.32%	73.02 ± 0.40%	82.72 ± 0.31%
NodeProperty	82.00 ± 0.82%	72.02 ± 0.31%	79.38 ± 0.47%
AGS (ours)	**83.56** ± 0.18%	**73.68** ± 0.20%	**83.02** ± 0.61%

The best result of each dataset is marked in bold.

## Data Availability

The data that support the findings of this study are openly available in https://github.com/kimiyoung/planetoid/raw/master/data.

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
