# Peer review of "Self-Supervised Node Classification with Strategy and Actively Selected Labeled Set"

_entropy, 2022, doi:10.3390/e25010030_

Round 1
Reviewer 1 Report
This paper aims to automatically design pretext task for self-supervised learning in GNN. The authors propose a bilateral learning framework which first determines the optimal pretext task with partial training data then trains the GNN model with selected pretext task. The authors also introduce active learning to select nodes to be labeled for efficiency. The node classification accuracy by the proposed framework shows improvement compared to the single pretext task setting.
Pros:
The bilateral optimizing framework incorporates all pretext task by weighted sum of loss of each task while training and selects the most effective pretext task based on validation set.
Ablation study on active learning strategy clearly supports the effectiveness of active learning that with less labeled node the trained model can achieve comparable results.
Adequate related works are discussed.
Cons:
The proposed learning framework is adapted from DARTS and the second order approximation is also the main contribution of DARTS. So the technical contribution is marginal.
The authors directly cite the experiment results reported in [1] while the authors change experiment settings to adapt to training process of the proposed framework. Such ambiguity in experiment setting leads to unfair comparison. A clear description of dataset splitting and experiment settings should be adopted. The baselines should run on the same data splitting settings for fair comparison.
Besides, there are some grammar errors, for example, line 7. There are some errors in equations. Some left brackets in Equation (12) are missing, and the notation in equation in line 190 cannot be found in context.
[1]Jin, W.; Derr, T.; Liu, H.; Wang, Y.; Wang, S.; Liu, Z.; Tang, J. Self-supervised learning on graphs: Deep insights and new direction. arXiv preprint arXiv:2006.10141 2020
In summary, the authors should revise the manuscript with respect to the description of experiment settings and run the baselines in the same settings. Besides, the English writing and mathematical equations should be carefully checked.
Reviewer 2 Report
This paper is about a fully-automatic methodology to do efficient semi-supervised learning (SSL) with the aim to proceed to node classification in graphs. For the details, see below: ** Global remarks: • The different state-of-the-Art’s seems complete and show a nice effort from the authors to provide all the information needed to show at the same time the novelty of their approach and to recall us all the elements necessary to understand the paper. • The two main contributions are : (1) the framework AGS which belongs to the family of auto ML techniques and which additionaly automatically optimizes the parameter « \beta » , to ensure that an optimal equilibrium is found between the two loss functions (the downstream/supervised one and the self-supervised one). This optimization allows us to reach a better combination of hyperparameters and weights for the final architecture optimization. -> The main difficulty of (1) comes from the fact that it is known that auto ML in self-supervised learning (SSL) is difficult (see the training and validation losses). (2) the active search with optimizes one step forward the previous approach, improving the provided method « AGS » into « AAGS » . • The ablation study is sufficient, and nicely shows the efficiency of the proposed active learning method added on AGS (except for the data set PubMed). ** Novelty: The works of this papers are clearly new and results from a smart combination of several modern and efficient methods (as exposed in the several sections of States-of-the-Art). Furthermore, (1) is an essential automatic procedure to reach a fully automatic method and is able to adapt itself to a given data set: It allows us also to adapt the numerical scheme to the graph data set distribution. This goal is nicely reached thanks to a second-order approximate differential-based optimization method (even if questionable, see below). ** In detail: • The English level really has to be improved : mainly, several phrases are cut or begin with words where the first letter is not in capital mode. Also « Faces this challenged » is repeated please remove an occurence. • why only 50 epochs? How can we justify this arbitrary choice? Would the results be better if the number of epochs is increased? Can we reach some overfitting as longly discussed in the introduction despite of the authors’ method? • does this methodology work on data set not related to graphs? Please discuss this point since here the experiments are limited to this case. • Can we really assume that a second-order is reasonable? Is it thanks to the approximations proposed in the paper? Indeed, it is generally not possible to get a differentiable at order one, so order two seems not realistic. Please discuss more this point. Indeed, results prove the efficiency of the method, but are some irregularities observed during the training? • At several places, I see the authors discuss about time consuming computations. Perhaps that it could be useful to add some graphics or some complexity computations of the current framework. • in the introduction, the dilemma about too « rich » labels and overfitting on one side, and the lack of robustness of supervised approaches to hacks on the other side are discussed. Please enter more into the details since it can be surprising for more that one reader that SSL methods can be so powerful, and perhaps detail a little more the explanations and intuitions provided in [5]? • why the functions applied on graph embedding are linear (end of page 5)? It seems too much restrictive. • In the same way, why using a GCN of order 2 for experiments? It seems too weak. It could be interesting to use several cases : {1,2,…,5}. • Please split the presented algorithm in several subparts, it is very difficult to read it this way. • There is a conflict between the set of beta’s that the authors call B, and the two notations B_e and B_l (same letters but different meanings). • Please detail much more the four last figures: please make explicit your observations and what do they mean according to you. ** Conclusion: there are many points to improve in this paper but, after modifications, it will deserve a publication in MDPI.Author Response
Please see the attachment

Round 2
Reviewer 1 Report
The authors have addressed all the issues that I mentioned in the previous version. I vote it for acceptance in current version.